# Experimental Study and Theoretical Analysis of Side-Pressure Laminated Bamboo Lumber Columns under Axial Compression

Shuai Liu [1], Danping Gao [1,2], Yazi Xie [1] and Bowang Chen [1,*]

1   School of Civil Engineering, Central South University of Forestry and Technology, Changsha 410004, China
2   School of Railway Construction, Hunan Technical College of Railway High-Speed, Hengyang 421002, China
*   Correspondence: tmcbw@163.com; Tel.: +86-15-6731-83820

**Abstract:** Side-pressure laminated bamboo lumber is made by gluing and pressing bamboo sheets together and can be used as a structural building material. The experiment and theoretical analysis are carried out for the side-pressure laminated bamboo lumber columns under axial compression in order to understand its performance under axial compression. In the experiment, the curve of load and lateral displacement in the middle of columns with different slenderness ratios is obtained under axial compression by considering the slenderness ratio (range: 23.1–92.4) of the specimen as a variable. Results show that the specimen undergoes an elastic stage, elastic-plastic stage, and failure stage when subjected to stress. The failure is characterized by a prominent ductility during this period. With an increase in the slenderness ratio, the elastic stage for the specimen is shortened, while the elastic-plastic stage is extended. Based on the geometric non-linear analysis, the pressure bar stability is analyzed for the specimen through the large deflection theory. A stable differential equation of the side-pressure laminated bamboo lumber column is established under axial compression. Based on the differential equation, the relationship between the bearing capacity of the axial center of the side-pressure laminated bamboo lumber column under axial compression and the lateral displacement in the middle of column can be derived as the reference for the application of side-pressure laminated bamboo lumber.

**Keywords:** side-pressure laminated bamboo lumber; axial compression; experimental study; pressure bar stability analysis; displacement calculation



## 1. Introduction

Brick and mortar structures are mainly adopted in the civil buildings in China, which causes severe environmental pollution, heavy carbon emissions, and low resource utilization [1]. Bamboo, as a renewable resource, is characterized by its short growth cycle, low energy consumption during processing, and convenient degradation after disposal. It has a tensile strength and a compressive strength about 2 times and 1.5 times that of wood, respectively, as well as a higher specific strength than wood and ordinary steel. Hence, bamboo is often praised by structural engineers as the "plant steel" and is an ideal green building material [2–5]. Bamboo is often referred to as raw bamboo, i.e., round bamboo or a bamboo slice that retains the initial properties of round lumber. However, raw bamboo can barely meet the requirement of modern structures and buildings due to its thin wall, cracks among each joint, small diameter, limited length, and variation of mechanical properties [6]. In recent years, raw bamboo has been further exploited and gradually formed into bamboo materials used in modern engineering, such as glued laminated bamboo, parallel strand bamboo, and laminated bamboo lumber [7–10]. Among them, the parallel strand bamboo is manufactured by compressing crushed bamboo bundle fibers into dense blocks with rectangular cross-sections (Figure 1) [11]. The laminated bamboo lumber is made by bamboo pieces with fixed width and thickness that have its outer skin and tabasheer removed. Then the bamboo pieces are dried to a moisture content of 8–12%

and are glued together by adhesive and pressing to form the laminated bamboo lumber. It is proved that the laminated bamboo lumber has high strength and can be used as a sort of construction material. It has broad prospects for application and has been adopted in floor, truck baseboard, furniture, and construction templates [12]. The laminated bamboo lumber can be classified into flat-pressure laminated bamboo lumber (Figure 2a), side-pressure laminated bamboo lumber (Figure 2b,d), and flat-side-pressure laminated bamboo lumber (Figure 2c). Side-pressure laminated bamboo has favorable mechanical properties and can be used in various structural members [13].

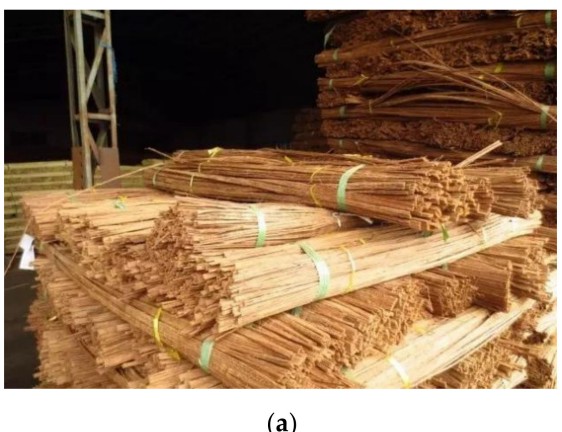

(a)

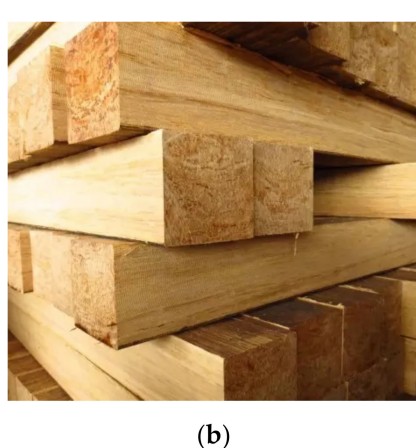

(b)

**Figure 1.** Production of parallel strand bamboo. (**a**) Bamboo bundle fiber. (**b**) Parallel strand bamboo.

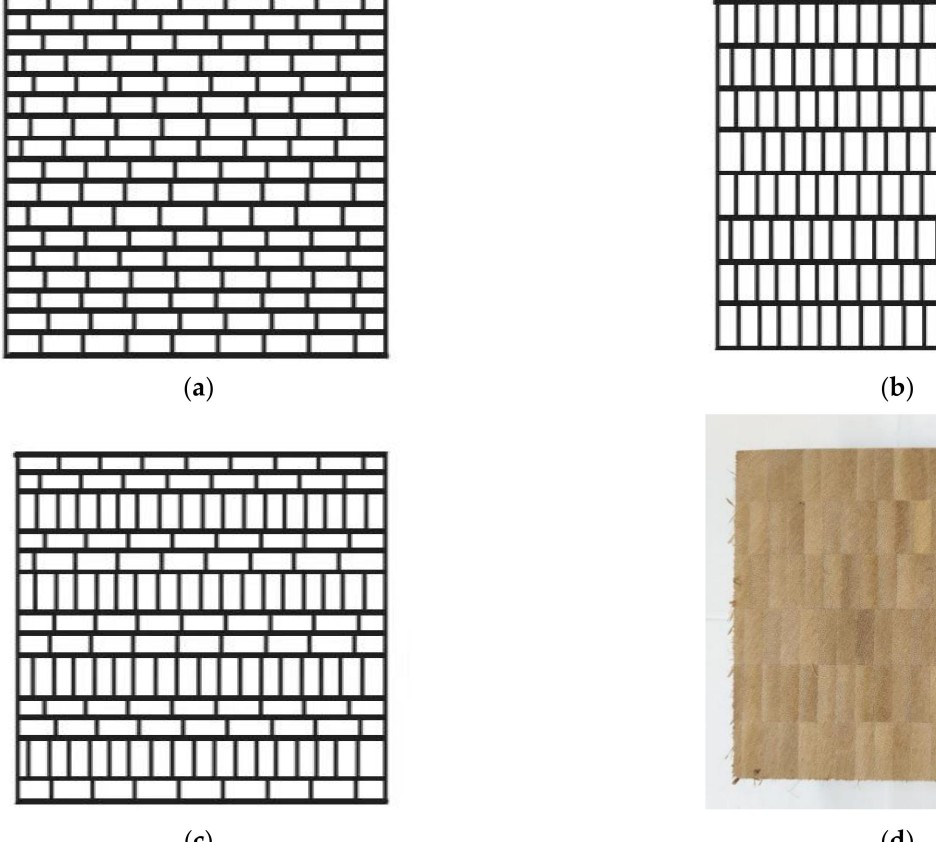

(a)

(b)

(c)

(d)

**Figure 2.** Classification of laminated bamboo lumber. (**a**) Flat-pressure laminated bamboo lumber. (**b**) Side-pressure laminated bamboo lumber. (**c**) Flat-side-pressure laminated bamboo lumber. (**d**) Side-pressure laminated bamboo lumber product.

In recent years, some scholars have conducted research on the mechanical properties of modern engineering bamboo materials and achieved positive results. Liu [14] conducted an axial compression study on parallel strand bamboo columns and proposed a corresponding equation for calculating the ultimate bearing capacity. The results suggest a good mechanical property and deformation capacity of parallel strand bamboo columns. Wang [15] conducted axial and eccentric compression tests on the parallel strand bamboo column, investigated its compression performance, damage mechanism, and damage pattern of the specimens, and proposed a nonlinear calculation method for the ultimate bearing capacity of its member with the rectangular section under compression-bending load. Tan et al. [16] conducted an experimental study and finite element analysis on the axial compression performance of parallel strand bamboo and side-pressure laminated bamboo lumber, proposed a corresponding calculation method for bearing capacity under axial compression, and derived the finite element analysis results. Li et al. [17,18] proposed the damage mechanism of side-pressure bamboo lumber by studying its compressive stress–strain model and eccentric compression. Su [19] obtained the compressive strength and elasticity modulus value in each direction by studying the mechanical properties of the square column pier of the side-pressure laminated bamboo lumber in three directions under axial compression. Xiao and Lv et al. [20,21] studied the glued laminated bamboo column under axial compression by considering the slenderness ratio as the variable, fitted the curve between the ultimate bearing capacity and the slenderness ratio of the bamboo columns, and proposed corresponding design suggestions. Tang et al. [22] investigated the long-term compressive performance of side-pressure laminated bamboo lumber. The results indicate that the ambient temperature and humidity had significant effects on the stress level.

The aforementioned studies related to modern engineering bamboo columns mainly focus on the parallel strand bamboo column rather than the side-pressure laminated bamboo lumber column. Thus, by considering the slenderness ratio as the variable, this study aims to provide reference for the research of side-pressure laminated bamboo lumber columns by investigating its mechanical properties and failure mode under axial compression, conducting pressure bar stability analysis on specimens through large deflection theory, and deriving the equation for calculating the bearing capacity of specimens and lateral displacement in the middle of the column.

## 2. Materials and Methods

### 2.1. Experimental Material and Fabrication of Specimens

The side-pressure laminated bamboo lumber used in the experiment was produced by Hunan Taohuajiang Bamboo Technology Company. The 3 or 5-year-old moso bamboo was processed into bamboo slices of about $8 \times 20$ mm in size. Then, the bamboo slices were treated with boiling water under a high temperature and carbonized under high pressure for 12 h, then pressed by structural adhesives to form the side-pressure laminated bamboo lumber. Its physical and mechanical properties were tested in accordance with test methods of national standard regarding wood density, moisture content, and parallel-to-grain compressive strength. The results are shown in Table 1.

**Table 1.** Properties of side-pressure laminated bamboo lumber.

| Property Index | Air-Dry Density | Moisture Content | Parallel-to-Grain Compression Strength |
|---|---|---|---|
| Result | 0.640 g/cm$^3$ | 6.3% | 84.9 MPa |
| Number of specimens | 32 | 32 | 32 |
| Standard Deviation | 0.04 | 0.003 | 3.04 |
| Coefficient of Variation | 0.06 | 0.04 | 0.04 |

In order to study the mechanical properties and failure mode of side-pressure laminated bamboo lumber columns with different slenderness ratios under axial compression, 15 specimens totaling 5 groups were designed in this experiment by taking slenderness ratio as the variable. Each specimen was 400 mm, 600 mm, 1000 mm, 1200 mm, and 1600 mm in length. The equations for calculating the slenderness ratio of specimens are as follow:

$$\lambda = \frac{l_0}{i} \qquad (1)$$

$$i = \sqrt{\frac{I}{A}} \qquad (2)$$

$$l_0 = k_l \times l \qquad (3)$$

where $\lambda$ is the slenderness ratio of the member under axial compression; $i$ is the turning radius of the cross-section of the member; $I$ is the inertia moment of the member; $A$ is the cross-sectional area of the member; $l$ is the actual length of the member; $l_0$ is the calculated length of the member; $k_l$ is the length calculation coefficient of the member. $k_l$ is taken as 1.0, as the member support is articulated at two ends.

The size of specimens is shown in Table 2.

**Table 2.** Parameters of each group specimen.

| Specimen No. | $l$/mm | $l_0$/mm | $\lambda$ | Size of Cross-Section (mm × mm) |
|---|---|---|---|---|
| A | 400 | 400 | 23.1 | 60 × 60 |
| B | 600 | 600 | 34.6 | 60 × 60 |
| C | 1000 | 1000 | 57.7 | 60 × 60 |
| D | 1200 | 1200 | 69.3 | 60 × 60 |
| E | 1600 | 1600 | 92.4 | 60 × 60 |

*2.2. Loading Regime and Arrangement of Measurement Points*

The loading regime in this experiment was carried out with reference to the Standard for Test Methods of Timber Structures (GB/T 50329-2012) [23]. Before the axial compression experiment was loaded, it was essential to pre-load all specimens in order to ensure that specimens were in good contact with the support. The pre-load value was taken as 1/50 of the estimated bearing capacity. The pre-load value was taken as 5 kN for specimens of group A and group B. The pre-load value was taken as 2 kN for specimens of group C, group D, and group E. The displacement control was adopted for the pre-loading of all specimens with a velocity of 1 mm/min. The loading got started after three pre-loadings were done completely. The loading process was divided into two stages, which were control of force value and displacement control. For the first stage, the loading speed of all specimens was obtained in accordance with the estimated ultimate bearing capacity to ensure that the peak loading could be realized between 5 min and 10 min as the loading began. Secondly, as the loading reached 80% of the pre-loading, it was shifted into displacement control loading. Then, the loading was terminated immediately and the residual force was discharged until the loading reduced to 80% of the peak load or brittle fracture occurred.

All specimens in this experiment were done in the laboratory of the School of Civil Engineering, Central South University of Forestry and Technology, using a 5000 KN microcomputer-controlled electro-hydraulic servo universal testing machine. The loading device for specimen A-1 is shown in Figure 3. According to the requirements of the Standard for Test Methods of Timber Structures (GB/T 50329-2012) on support device in the axial compression experiment, a bi-directional support was used as the support device to ensure that the specimen could be rotated around any axis in two mutually perpendicular axes of the section, as shown in Figure 4.

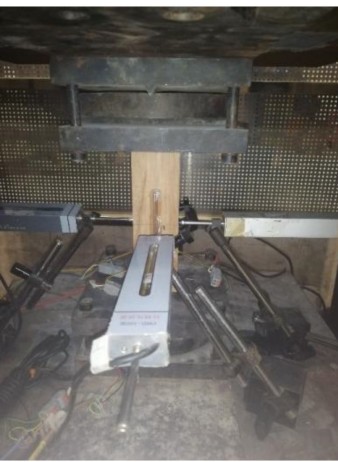

**Figure 3.** Specimen-loading device.

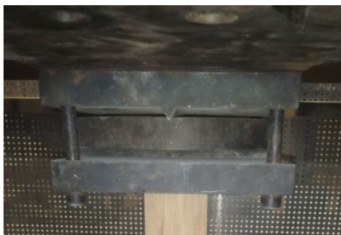

**Figure 4.** Bi-directional support.

This experiment aimed to measure: (1) the bearing capacity, which was automatically collected by the servo universal testing machine; (2) the lateral displacement of the specimen, which was collected by a YWD-100-type (maximum range: ±100 mm) displacement transducer connected to the DH3818 static strain test system.

The displacement meters were arranged in the middle of each side of the specimen, with four meters in total (LVDT 1 to 4). The measurement points of the specimen were arranged as shown in Figure 5.

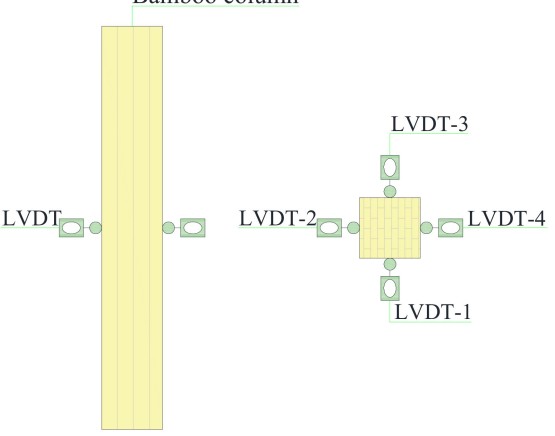

**Figure 5.** Displacement meter arrangement of the specimens (left: front view, right: top view).

## 3. Experimental Results and Discussion

### 3.1. Experimental Result and Failure Characteristics

According to the Standard for Design of Timber Structures (GB/T 50005-2017) [24], specimens with a height–width ratio no greater than 10 or a slenderness ratio no greater than 35 are short columns, and specimens with a height–width ratio greater than 10 or

a slenderness ratio greater than 35 are long columns. During the experiment, the short-column specimens in group A and group B are subjected to strength failure because the load reaches the strength limit. The long-column specimens in group C to group E are subjected to the buckling failure due to excessive lateral displacement. Table 3 shows the axial compression load and lateral displacement data in the middle of the specimens. Among them, the lateral displacement data of specimen B-3 and specimen C-3 are lost due to the failure of the data acquisition system during the experiment.

**Table 3.** Ultimate load and maximum lateral displacement of specimens.

| Specimen No. | $\lambda$ | $N_{\text{Peak,i}}$/kN | $N_{\text{Peak}}$/kN | $y_{\text{max,i}}$/mm | $y_{\text{max}}$/mm |
|---|---|---|---|---|---|
| A-1 | 23.1 | 244.4 |  | 10.04 |  |
| A-2 | 23.1 | 245.7 | 245.8 | 9.65 | 9.92 |
| A-3 | 23.1 | 247.3 |  | 10.07 |  |
| B-1 | 34.6 | 144.1 |  | 23.84 |  |
| B-2 | 34.6 | 221.9 | 188.9 | 16.7 | 20.27 |
| B-3 | 34.6 | 200.9 |  | — |  |
| C-1 | 57.7 | 92.7 |  | 28.95 |  |
| C-2 | 57.7 | 96.6 | 91.7 | 31.61 | 30.28 |
| C-3 | 57.7 | 86.0 |  | — |  |
| D-1 | 69.3 | 71.9 |  | 36.19 |  |
| D-2 | 69.3 | 80.0 | 73.8 | 42.74 | 38.93 |
| D-3 | 69.3 | 69.5 |  | 37.86 |  |
| E-1 | 92.4 | 58.6 |  | 61.16 |  |
| E-2 | 92.4 | 56.4 | 57.6 | 66.51 | 64.14 |
| E-3 | 92.4 | 57.8 |  | 64.74 |  |

Note: $\lambda$ is the slenderness ratio; $N_{\text{Peak,i}}$ is the measured value of peak load of each specimen; $N_{\text{Peak}}$ is the average value of peak load of each group of specimens; $y_{\text{max,i}}$ is the maximum value of lateral displacement in the middle of each specimen; $y_{\text{max}}$ is the average value of the maximum value of lateral displacement in the middle of each group of specimens.

Strength failure is the failure mode of specimen A and specimen B. Taking specimen A-1 as an example, in the early stage of loading, specimen A-1 exhibits an increasing load and unapparent lateral deformation. When the load rises to 80–90% of the peak load, the specimen enters the plastic stage. After the peak load is reached, the load begins to decrease slowly. When the load falls to about 90% of the peak load, the specimen emits a slight fiber fracture sound. Then, the fracture sound gets increasingly intensive. The specimen shows an increasing lateral deformation, with the column bent. At last, the bamboo piece in the middle of the tensile side of the specimen is fractured. In addition, the bamboo piece in the compressive side is bent, resulting in strength failure. Figure 6 shows the failure mode of specimen A-1. The remaining specimens in group A and group B have similar failure modes.

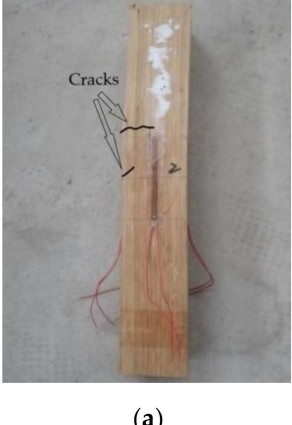
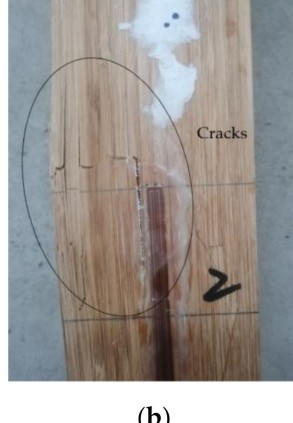

(**a**)　　　　　　　　　　　　　　　　　　(**b**)

**Figure 6.** Failure mode of the A-1 specimen. (**a**) Overall image. (**b**) Partial image.

Buckling failure is the failure mode of the specimens from group C to group E. Taking specimen C-1 as an example, in the early stage of loading, the specimen exhibits an increasing load with unapparent lateral deformation, similar to group B. When the load rises to about 60% to 80% of the peak load, the specimen enters the plastic stage with a significant lateral deformation. When the peak load is reached, the specimen emits a fiber fracture sound. With increasing lateral displacement, the specimen is destabilized. Then, the load gradually decreases until the experiment is done. The column can be observed with a salient bending for its shape, with a fracture occurring at the middle. The deformation significantly decreases after the specimen is unloaded at the end of the experiment. Then, less residual deformation is observed, and the buckling failure occurs. Figure 7 shows the failure mode of specimen C-1. The failure mode of group D and the rest of the specimens in group C are similar to specimen C-1. The fracture of specimens in group D also occurs in the middle of the column and has a more prominent failure mode than that in group C. As shown in Figure 8, the fracture expands along the corner on two cross-sections. The damage process of specimens in group E is similar to that of specimen C-1. However, obvious cracks are barely observed on the specimen surface and no obvious cracking sound is heard during loading.

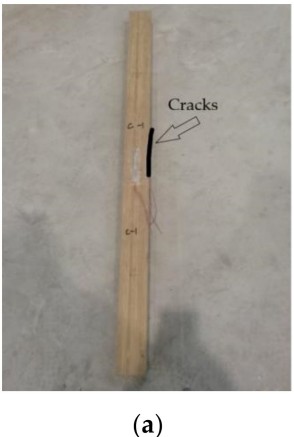
(a)

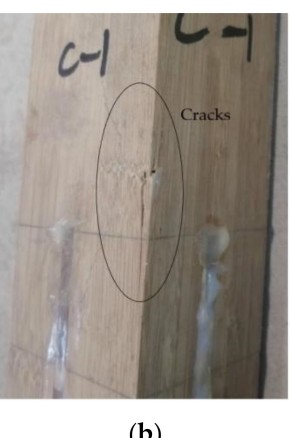
(b)

**Figure 7.** Failure mode of the C-1 specimen. (**a**) Overall image. (**b**) Partial image.

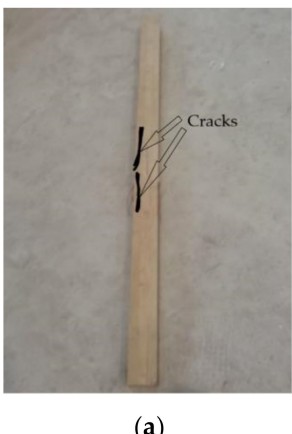
(a)

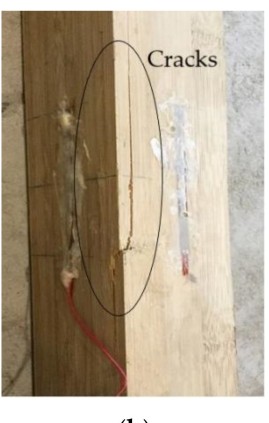
(b)

**Figure 8.** Failure mode of the D-1 specimen. (**a**) Overall image. (**b**) Partial image.

*3.2. Load-Lateral Displacement Curve*

Figure 9 shows the load-lateral displacement curve of the middle of the column in each group specimen. As observed in Figure 9, the load capacity of the specimen is weakened, and lateral deformation is salient with a larger slenderness ratio. The specimen undergoes an elastic stage, elastic-plastic stage, and failure stage under stress. The short-column

specimens in group A and group B are in the elastic stage, with slight lateral displacement from the beginning of loading to the time when the load is about to peak. When the loading reaches around 80% to 90% of the peak load, the specimen enters an elastic-plastic stage, with increasing lateral displacement in the middle of column until the strength failure occurs when the bamboo fiber of the tension face is snapped and that of the compression face is buckled. The long-column specimens of group C to group E are in the elastic stage during the beginning of loading, with unapparent lateral deformation. When the loading reaches around 50% to 60% of the peak load, the specimen enters an elastic-plastic stage, which is faster than the short column. With the increasing load, the lateral displacement in the middle of column increases until buckling failure occurs on the specimen due to bending.

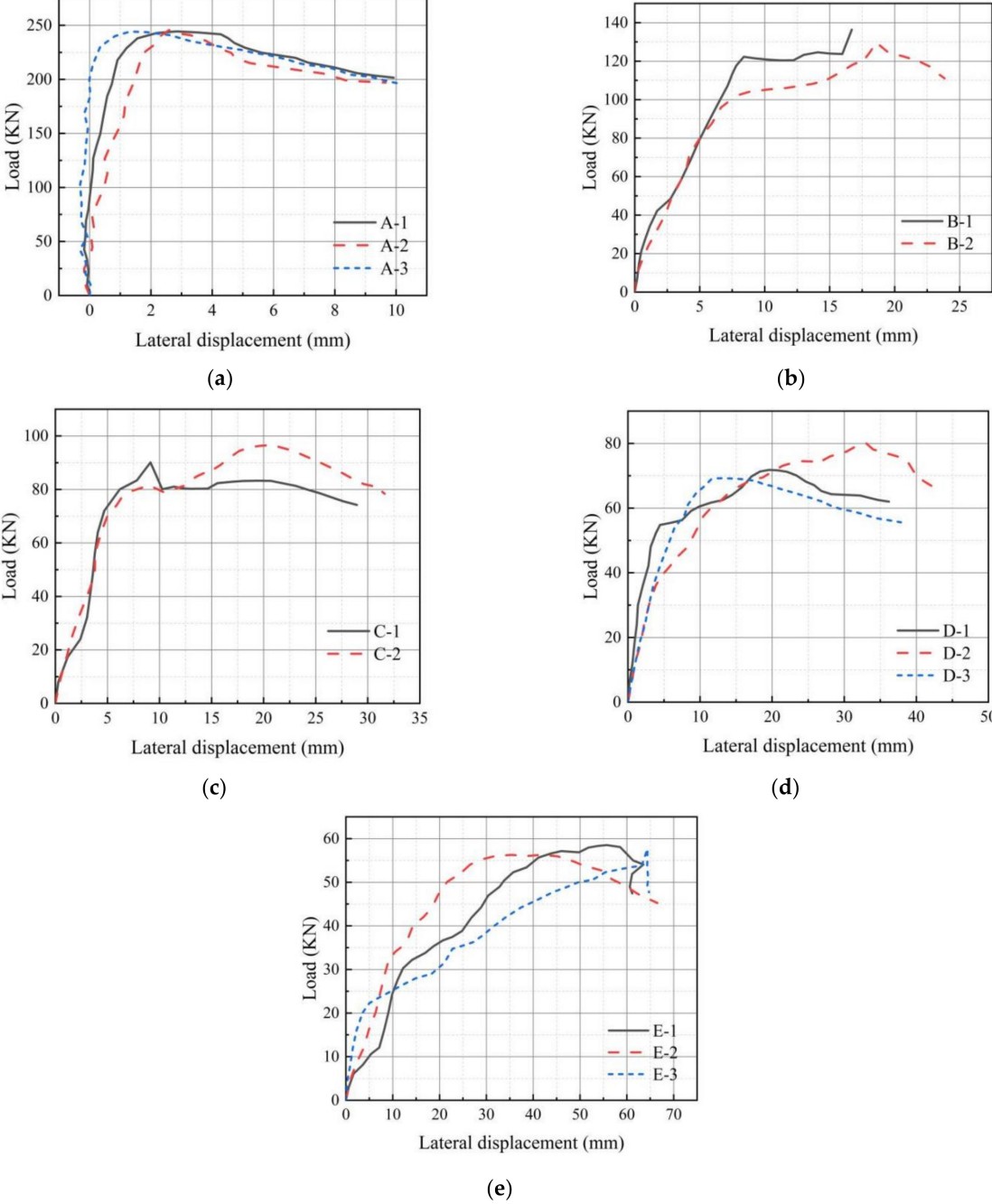

**Figure 9.** Load-lateral displacement curve in the middle of the column. (**a**) Group A. (**b**) Group B. (**c**) Group C. (**d**) Group D. (**e**) Group E.

With an increasing slenderness ratio of the specimen, the elastic deformation stage for each group of specimens is gradually shortened, and the specimens in group E that have the largest slenderness ratio do not exhibit a prominent elastic deformation. The specimen E-1 is initially subject to a certain bending before the test. Its elastic deformation is shorter and more obvious than the other two specimens in the same group during the loading process. In addition, its final bending direction is the same as the initial bending direction. This suggests that long columns are more easily affected by the initial eccentricity and material defects and enter the plastic deformation stage in advance.

Figure 10 shows the curve describing the relationship between the lateral displacement and the slenderness ratio in the middle of the column. As the slenderness ratio increases, the flexure second-order deformation caused by the specimen under axial compression is increasingly significant. At the time of damage, the column suffers greater from deformation when bending. For the specimen with a smaller slenderness ratio, the lateral displacement is small when the damage occurs. The specimen is deprived of its load-bearing capacity when the bamboo fibers are crushed or pulled off. In this case, the failure mode is strength failure. After peak loading is reached, the specimens with a greater slenderness ratio gradually suffer from buckling failure, because the balance can hardly be kept on specimens that bear axial compression and an additional bending moment simultaneously. Moreover, with a larger slenderness ratio, the lateral deformation is reached more quickly after buckling failure. In this case, the failure mode is buckling failure. The average lateral displacement of specimens in groups B to group E increases by about 1, 2, 3, and 5.5 times compared with group A, respectively.

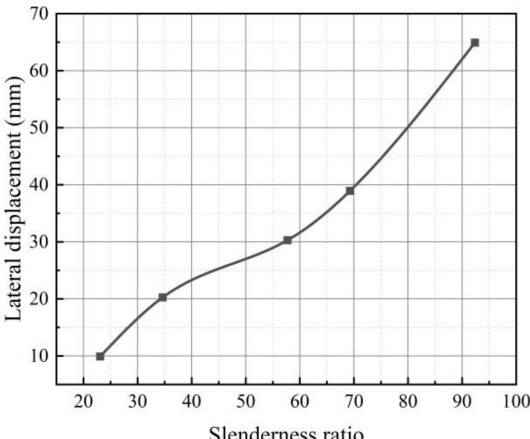

**Figure 10.** Slenderness ratio–lateral displacement curve.

## 4. Analysis of Column Load and Lateral Displacement under Axial Compression

### 4.1. Analysis of Rod Stability under Axial Compression

The analysis of rod stability under axial compression includes small deflection theory and large deflection theory. The small deflection theory is applicable to the case where only a small portion of deformation occurs in the compressed rod. The original size can be used as the basis of calculation. The large deflection theory is applicable to the case where excessive deformation occurs in the compressed rod. The size after deformation can be used as the basis of calculation. The nonlinear problems caused by large deformation are referred to as geometric nonlinear analysis [25]. Side-pressure laminated bamboo lumber is an anisotropic material characterized by prominent nonlinearity in terms of the mechanical property. Thus, it is accurate to analyze the relationship between load and deformation of the rod under axial compression and buckling through large deflection theory.

### 4.2. Stable Differential Equation

The deformation of the column under axial compression is shown in Figure 11, where s represents the arc length of the deflection curve.

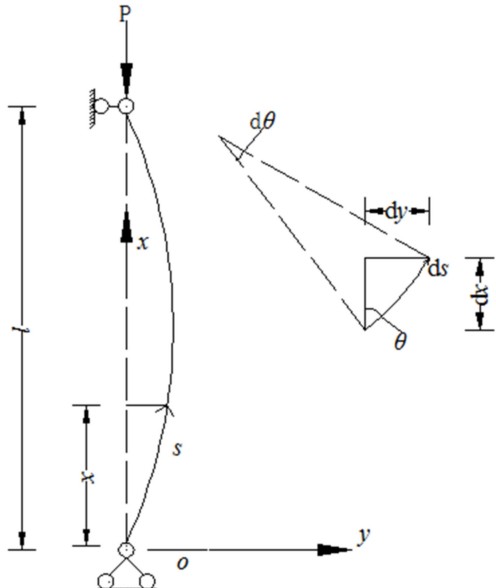

**Figure 11.** Deformation of column under axial compression.

The following is obtained based on the physical conditions:

$$\frac{1}{\rho} = \frac{M}{\beta EI} \tag{4}$$

where $\rho$ is the radius of curvature on the deflection curve with arc length $s$; $\beta$ is the reduction factor of the elasticity modulus for the nonlinear deformation of the material; the error between the theoretical result and the experimental result is calculated to be the slightest when $\beta$ is 0.62, according to the measured value of lateral displacement under axial compression. Hence, $\beta$ is taken as 0.62.

Equation (5) is obtained as follows based on geometric conditions:

$$\frac{1}{\rho} = -\frac{d\theta}{ds} \tag{5}$$

The Equation (6) is obtained as follows by substituting Equation (4) into Equation (5):

$$\frac{d\theta}{ds} = -\frac{M(s)}{\beta EI} \tag{6}$$

The following equation is obtained by substituting $M(s) = Py(s)$ into Equation (6):

$$\beta EI\frac{d\theta}{ds} + Py(s) = 0 \tag{7}$$

Equation (8) is obtained by substituting $\alpha^2 = P/EI$ into Equation (7):

$$\beta\frac{d\theta}{ds} + \alpha^2 y(s) = 0 \tag{8}$$

The following is obtained by calculating the derivative of $s$ in Equation (8):

$$\beta\frac{d^2\theta}{ds^2} + \alpha^2\frac{dy(s)}{ds} = 0 \tag{9}$$

Substituting $\mathrm{d}y/\mathrm{d}s = \sin\theta$ into Equation (9), we can obtain:

$$\beta\frac{\mathrm{d}^2\theta}{\mathrm{d}s^2} + \alpha^2\sin\theta = 0 \tag{10}$$

Equation (10) is the stable differential equation of large deflection theory.

### 4.3. Lateral Displacement Solution

To investigate the relationship between load-bearing capacity and lateral displacement of middle of the column, the stable differential equation is solved.

If $t = \mathrm{d}\theta/\mathrm{d}s$, we have:

$$\frac{\mathrm{d}^2\theta}{\mathrm{d}s^2} = \frac{\mathrm{d}t}{\mathrm{d}s} = \frac{\mathrm{d}t}{\mathrm{d}\theta}\cdot\frac{\mathrm{d}\theta}{\mathrm{d}s} = t\frac{\mathrm{d}t}{\mathrm{d}\theta} \tag{11}$$

Equation (12) is obtained by substituting Equation (11) into Equation (10):

$$\beta t\frac{\mathrm{d}t}{\mathrm{d}\theta} + \alpha^2\sin\theta = 0 \tag{12}$$

Equation (13) is obtained by separating the variables and through integration:

$$\beta t^2 - 2\alpha^2\cos\theta = C \tag{13}$$

where $C$ is an undefined constant.

Based on conditions of the rod end, when $s = 0$, $\theta = \theta_0$ and $M(o) = 0$, we have:

$$M(o) = -\beta EI\left(\frac{\mathrm{d}\theta}{\mathrm{d}s}\right)_{s=0} = -\beta EI(t)_{s=0} = 0 \tag{14}$$

i.e., $(t)_{s=0} = 0$. Equation (15) is obtained by substituting $(t)_{s=0} = 0$ into Equation (13):

$$C = -2\alpha^2\cos\theta_0 \tag{15}$$

By substituting Equation (15) into Equation (13), Equation (16) is obtained after the square root:

$$t = -\alpha\sqrt{\frac{2(\cos\theta - \cos\theta_0)}{\beta}} \tag{16}$$

By substituting $t = \mathrm{d}\theta/\mathrm{d}s$, $\cos\theta = 1 - 2\sin2(\theta/2)$, and $\cos\theta_0 = 1 - 2\sin2(\theta_0/2)$ into Equation (16) and integrating s from 0 to $l/2$ and $\theta$ from 0 to $\theta_0$, we can obtain the Equation as follows:

$$\alpha l = \sqrt{\beta}\int_0^{\theta_0}\frac{\mathrm{d}\theta}{\sqrt{\sin^2\frac{\theta_0}{2} - \sin^2\frac{\theta}{2}}} \tag{17}$$

For the convenience of calculation, Equation (17) is transformed into an elliptic integral with $\sin(\theta_0/2) = P$ and $\sin(\theta/2) = P\sin\varphi$. Therefore, we have:

$$\mathrm{d}\theta = \frac{2P\cos\varphi}{\sqrt{\beta(1 - P^2\sin\varphi)}}\mathrm{d}\varphi \tag{18}$$

When $\theta = 0$, $\varphi = 0$. When $\theta = \theta_0$, $\varphi = \pi/2$. Therefore, we have:

$$\frac{\alpha l}{2} = \int_0^{\frac{\pi}{2}}\frac{1}{\sqrt{\beta(1 - P^2\sin\varphi)}}\mathrm{d}\varphi \tag{19}$$

The right side of Equation (19) is complete elliptic integrals of the first kind, whose values are related to $P = \sin(\theta_0/2)$, which can be found by the table of elliptic integrals [26].

Based on $\mathrm{d}y = \sin\theta\mathrm{d}s$, Equation (20) is obtained with reference to Equation (16):

$$\mathrm{d}y = -\frac{2q\sqrt{\beta}}{\alpha}\sin\varphi\mathrm{d}\varphi \tag{20}$$

Equation (21) is obtained through integration.

$$y_{\text{theory}} = -\frac{2P\sqrt{\beta}}{\alpha}\int_{\frac{\pi}{2}}^{\varphi}\sin\varphi\mathrm{d}\varphi = \frac{2P\sqrt{\beta}}{\alpha}\cos\varphi \tag{21}$$

Equation (21) describes the relationship between the load-bearing capacity and lateral displacement of the column under axial compression.

The maximum value of lateral displacement occurs at the middle of the compressed column, i.e., $\theta = 0$. Correspondingly, $\varphi = 0$. Therefore, we can obtain:

$$y_{\text{max,theory}} = \frac{2P\sqrt{\beta}}{\alpha} = \frac{Pl\sqrt{\beta}}{\left(\frac{\alpha l}{2}\right)} \tag{22}$$

Equation (22) is the theoretical expression for the maximum lateral displacement of the column under axial compression.

### 4.4. Comparison between Theoretical Value and Experimental Value

The theoretical value of the maximum lateral displacement of the side-pressure laminated bamboo lumber column under axial compression calculated by Equation (22) is compared with the actual value. The comparison results are shown in Table 4.

**Table 4.** Result of comparison for lateral displacement in the middle of the column.

| Specimen No. | $l$/mm | $\lambda$ | $\alpha l/2$ | $y_{\text{max, real}}$ | $y_{\text{max, theory}}$ | Error (%) |
|---|---|---|---|---|---|---|
| A | 400 | 23.1 | 1.5718 | 9.92 | 9.94 | 0.20% |
| B | 600 | 34.6 | 1.5726 | 20.27 | 20.30 | 0.14% |
| C | 1000 | 57.7 | 1.5722 | 30.28 | 30.33 | 0.17% |
| D | 1200 | 69.3 | 1.5725 | 38.93 | 38.99 | 0.15% |
| E | 1600 | 92.4 | 1.5733 | 64.14 | 64.20 | 0.10% |

Note: In the table, $y_{\text{max, real}}$ is the average value of the measured maximum value of lateral displacement in each group of specimens; $y_{\text{max, theory}}$ is the average value of the theoretical maximum value of lateral displacement in each group of specimens.

It can be observed from Equation (19), the $P$ in Equation (22) is only related to the turning angle $\theta_0$ at both ends of the specimen. The device for measuring the turning angle at both ends is not set in the experiment. Thus, $\theta_0$ is estimated according to the actual measured value of lateral displacement in the column. The table shows that the error between the experimental result and the theoretical result is not more than 1%. Equation (22) can reflect the relationship between the bearing capacity and the lateral displacement in the middle of the column under axial compression.

## 5. Conclusions

(1) The mechanical property of side-pressure laminated bamboo lumber under axial compression is studied with reference to the Standard for Test Methods of Timber Structures (GB/T 50329-2012). The study shows that short-column specimens in group A and group B are subjected to strength failure because the bamboo fibers on the tensile side are pulled off and those on the compressive side are flexed. The long-column specimens in group C to group E undergo buckling failure due to excessive lateral displacement during the experiment.

(2) The side-pressure laminated bamboo lumber column undergoes an elastic stage, elastic-plastic stage, and failure stage under axial compression in that order. The

damage is characterized by obvious ductility. With the increase of the slenderness ratio, the elastic stage is gradually shortened, while the elastic-plastic stage is prolonged.

(3) As an anisotropic material, the side-pressure laminated bamboo lumber has prominent nonlinearity in terms of the mechanical properties. The material is subjected to deformation under compression. Thus, large deflection theory should be used as the reference to analyze the rod under axial compression.

(4) The stable differential equation is used to solve the established side-pressure laminated bamboo lumber column. The relationship between the ultimate bearing capacity of the compressed column and the lateral displacement in the middle of the column is derived. The error between the theoretical and experimental values is not greater than 1%, with high goodness of fit. It is hoped that this research can provide an effective reference for future engineering applications.

**Author Contributions:** Conceptualization, B.C. and S.L.; Methodology, B.C. and S.L.; Software, S.L.; Validation, D.G. and Y.X.; Formal analysis, S.L.; Investigation, B.C.; Resources, B.C.; Data curation, S.L.; Writing—original draft preparation, S.L.; Writing—review and editing, B.C.; Visualization, D.G. and Y.X.; Supervision, B.C.; Project administration, B.C.; Funding acquisition, B.C. All authors have read and agreed to the published version of the manuscript.

**Funding:** This research was funded by Hunan Province High-tech Industry Science and Technology Innovation Leading Program, grant number 2022GK4052.

**Institutional Review Board Statement:** Not applicable.

**Informed Consent Statement:** Not applicable.

**Data Availability Statement:** All data, models, and code generated or used during the study appear in the submitted article.

**Conflicts of Interest:** The authors declare no conflict of interest.

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
