# Peer review of "Experimental Study and Theoretical Analysis of Side-Pressure Laminated Bamboo Lumber Columns under Axial Compression"

_sustainability, doi:10.3390/su141811360_

Round 1

Reviewer 1 Report

Comments sustainability-1889421

The manuscript entitled " Experimental Study and Theoretical Analysis of Side-Pressure Laminated Bamboo Lumber Column under Axial Compression" studied the side-pressure laminated bamboo lumber column under axial compression. The research topic is exciting and essential for the promotion of bamboo structures; the reviewer suggests the manuscript be accepted for publication as a technical paper after addressing a few comments:

Line 12: the range of the ratio should be given;

Line 75-76: the fundamental differences between the "parallel strand bamboo" and "side-pressure laminated bamboo" should be concisely explained here;

Line 85 …WAS produced…, the tense of this manuscript should be carefully checked;

Line 88-89: the St. D of mechanical properties should be given;

Line 102: should be estimated bearing capacity;

Overall, the induced shear stress (similar to roll-shear in beam components) significantly influences buckling performance, and this point should be considered in the analysis part. 

Author Response

Response to Reviewer Comments

 Please refer to the attachment for the revised manuscript.

Comment: The manuscript entitled " Experimental Study and Theoretical Analysis of Side-Pressure Laminated Bamboo Lumber Column under Axial Compression" studied the side-pressure laminated bamboo lumber column under axial compression. The research topic is exciting and essential for the promotion of bamboo structures; the reviewer suggests the manuscript be accepted for publication as a technical paper after addressing a few comments.

Response: Thank you very much for your interest and recognition of our work, and for your careful and constructive comments on our manuscript. We have revised this paper in accordance with your comments. Here we provide an item-by-item response to your comments.

Point 1: Line 12: the range of the ratio should be given.

Response 1: Thank you very much for your suggestions. We have added the range of the ratio as required, please refer to line 14 of our manuscript.

Point 2: Line 75-76: the fundamental differences between the ”parallel strand bamboo” and “side-pressure laminated bamboo” should be concisely explained here.

Response 2: Thank you very much for your suggestions. For the differences between the ”parallel strand bamboo” and “side-pressure laminated bamboo” we have given additional explanations in lines 42-48. We also give pictures of both, please refer to Figure 1, Figure 2.

Point 3: Line 85 …WAS produced…, the tense of this manuscript should be carefully checked.

Response 3: We appreciate your pointing out the errors in this manuscript. We have carefully checked the tense of this manuscript and corrected some errors.

Point 4: Line 88-89: the St. D of mechanical properties should be given.

Response 4: Thank you very much for your suggestion, we have provided the standard deviation of the mechanical properties as requested and have provided the number of specimens and coefficient of variation of the material properties tests, please refer to Table 1 of the manuscript for this section.

Point 5: Line 102: should be estimated bearing capacity.

Response 5: We appreciate your pointing out the errors in this manuscript. The text has been revised as required. Please refer to the line 118 in our manuscript.

Point 6: Overall, the induced shear stress (similar to roll-shear in beam components) significantly influences buckling performance, and this point should be considered in the analysis part. 

Response 6: Thank you very much for your comments. In lattice columns, the induced shear stress has a significant effect on the buckling performance. And in solid web columns, the shear stress has less impact on the buckling performance. Therefore, the effect of induced shear stress on the buckling performance of the side-pressure laminated bamboo lumber solid web columns is not considered in this test.

Reviewer 2 Report

Title: Experimental Study and Theoretical Analysis of Side-Pressure Laminated Bamboo Lumber Column under Axial Compression

Sustainable Development of Timber Buildings:

Submitted to section: Green Building,

 Comments:

General:

The study was interesting and add theoretical knowledge for engineering a bamboo column from side-pressure laminated bamboo lumber by considering the slenderness ratio as the variable. The mechanical property and failure mode under axial compression was thoroughly discussed alongside with the derivation of equation for calculating the bearing capacity of specimens and lateral displacement in the middle of the column.

My comments, however, was related to the property of the bamboo species used in making the side-pressure laminated bamboo lumber as well as the adhesive used in lamination. Since, bamboo species, like wood species, vary in their physical and mechanical properties, it is important that it should also be considered in the design aspect when engineering a bamboo column. The same is true for the adhesive since there are structural and non-structural adhesives.  Hence, I would suggest that these two factors should also be given consideration in the engineering design of a bamboo column.

Specific:

Introduction:

Please add some additional reviews on the effect of bamboo species,  adhesives, and treatment (i.e, chemical preservation or carbonization) on bamboo column testing/performance as part of the introduction.

Materials and Methods

Ø  Please provide an illustration of side-pressure laminated bamboo lumber for clarity as not all potential readers of this paper can understand the bamboo lamination method done.

Ø  Line  84 – 89 - The specifications given by the manufacturer were lacking. It is important that bamboo species used are mentioned as well as the adhesive used in lamination and the preservative treatment done on bamboo poles (to prevent powderpost beetles' attack). Also, it is important to know if the bamboo slats/sheets used were carbonized and at what temperature (if possible to be obtained from the source)?

Ø  Line 90 – 93 – The sample size or replicates used were only 3 per group. As bamboo like wood is a very variable material especially that properties varied along the length, within bamboo clumps, or among bamboos from the same species, sample size should have been more than 3 to increase statistical reliability of the results. In case of data loss as what happened with B3 and C3, they could be replaced by other samples and outlier data can be removed in the analysis (if necessary). Hence, I am not so convinced in the results with very few samples/replicates.

Author Response

Response to Reviewer Comments

Please refer to the attachment for the revised manuscript. 

Comment: The study was interesting and add theoretical knowledge for engineering a bamboo column from side-pressure laminated bamboo lumber by considering the slenderness ratio as the variable. The mechanical property and failure mode under axial compression was thoroughly discussed alongside with the derivation of equation for calculating the bearing capacity of specimens and lateral displacement in the middle of the column.

My comments, however, was related to the property of the bamboo species used in making the side-pressure laminated bamboo lumber as well as the adhesive used in lamination. Since, bamboo species, like wood species, vary in their physical and mechanical properties, it is important that it should also be considered in the design aspect when engineering a bamboo column. The same is true for the adhesive since there are structural and non-structural adhesives.  Hence, I would suggest that these two factors should also be given consideration in the engineering design of a bamboo column.

Response: Thank you very much for your interest and recognition of our work, and for your careful and constructive comments on our manuscript. We have revised this paper in accordance with your comments. Here we provide an item-by-item response to your comments.

Point 1: Please add some additional reviews on the effect of bamboo species,  adhesives, and treatment (i.e, chemical preservation or carbonization) on bamboo column testing/performance as part of the introduction.

Response 1: Thank you very much for your suggestion. The Chinese standard GB/T 40487-2021, Structural Bamboo Glulam specifies the manufacturing requirements of a laminated bamboo, including the use of adhesive. In addition, the side-pressure laminated bamboo lumber in this study is a modern engineered bamboo material made of 3 or 5-year-old moso bamboo slices. The bamboo slices were subjected to 12 hours of high temperature boiling and high pressure carbonization, and then were produced through specific processes such as gluing and hot pressing. Hence, the side-pressure laminated bamboo lumber has less variability than wood and natural bamboo. A study (https://doi.org/10.1016/j.conbuildmat.2017.06.005) shows that a total number of 62 or 30 clear specimens provided similar statistical results. A study (10.19701/j.jzjg.20200631) shows that material damage occurs in the side-pressure laminated bamboo under compression and no damage occurs in the adhesive layer. In this paper, we mainly study the effect of slenderness ratio on the axial compression performance of side-pressure laminated bamboo lumber columns, without considering bamboo species, adhesive and treatment (i.e, chemical preservation or carbonization) as the main influencing factors. However, it is undeniable that your suggestion is a very good research direction, which provides a new idea for our subsequent research on bamboo integrates. Thank you again.

Point 2:  Please provide an illustration of side-pressure laminated bamboo lumber for clarity as not all potential readers of this paper can understand the bamboo lamination method done.

Response 2: We appreciate your comments and suggestions. Please refer to lines 42-48, 92-97, and Figure 2 of the manuscript for a description of this section.

Point 3: Line  84 – 89 - The specifications given by the manufacturer were lacking. It is important that bamboo species used are mentioned as well as the adhesive used in lamination and the preservative treatment done on bamboo poles (to prevent powderpost beetles' attack). Also, it is important to know if the bamboo slats/sheets used were carbonized and at what temperature (if possible to be obtained from the source)?

Response 3: Thank you for your comments and suggestions, we have added the relevant content as requested, please refer to lines 92-97.

Point 4:   Line 90 – 93 – The sample size or replicates used were only 3 per group. As bamboo like wood is a very variable material especially that properties varied along the length, within bamboo clumps, or among bamboos from the same species, sample size should have been more than 3 to increase statistical reliability of the results. In case of data loss as what happened with B3 and C3, they could be replaced by other samples and outlier data can be removed in the analysis (if necessary). Hence, I am not so convinced in the results with very few samples/replicates.

Response 4: Thank you very much for your comments, and we also agree with you that this is one of the limitations of this study. However, side-pressure laminated bamboo lumber has less variability than wood and natural bamboo. And the results of the material property tests (Table 1) also prove this feature of small variability of the side-pressure laminated bamboo lumber. In addition, considering the test cost, there are only three samples per group of specimens in this study.

Reviewer 3 Report

The paper presents an experimental analysis of bamboo timber pressed laterally. The paper is direct and the results are fairly interesting and supported by the data. I would recommend better explaining the proposed methodology and in the last paragraph of the introduction, giving the reader an idea of what to expect from all the sections of the paper.

Author Response

Response to Reviewer Comments

Please refer to the attachment for the revised manuscript.

Comment: The paper presents an experimental analysis of bamboo timber pressed laterally. The paper is direct and the results are fairly interesting and supported by the data. I would recommend better explaining the proposed methodology and in the last paragraph of the introduction, giving the reader an idea of what to expect from all the sections of the paper.

Response: Thank you very much for your interest and recognition of our work. Thank you very much for your suggestions as well, we have already explained the work to be done in this manuscript in the introduction section.

Reviewer 4 Report

Experimental Study and Theoretical Analysis of Side-Pressure Laminated Bamboo Lumber Column under Axial Compression

The possibility of using bamboo-based beams as construction materials is an interesting direction in the development of wooden buildings. An important issue in this case is the strength characteristics of bamboo beams. In this context, the topic of the article should be considered interesting and topical. The introduction and the literature review contained in it are substantively prepared and well introduce the reader to the subject of the article. The cited literature is up-to-date, mostly from the last few years. The work methodology is generally well described. Nevertheless, authors should better characterize the research material. The authors of the study used side-pressure laminated bamboo lumber column. In the literature review, they state that so far research has been conducted rather on the basis of the parallel strand bamboo column. The authors should explain in more detail what is the difference between side-pressure laminated bamboo lumber column and parallel strand bamboo column. Perhaps it would be helpful to take a picture of the cross-section of the tested material.

Why did the authors use only 3 samples for each variant (additionally, a system failure excluded 1 sample from variants B and C). As a standard, in strength tests, approx. 10 samples per variant are used.

With regard to formula 2, there is no explanation of what the symbol "A" means.

The results of experimental research should be confronted by the authors with the data available in the literature. In the entire analysis of the results, there is no reference to the results of studies carried out by other researchers.

Taking into account the above remarks, the article requires a minor revision.

Author Response

Response to Reviewer Comments

Please refer to the attachment for the revised manuscript.

Comment: The possibility of using bamboo-based beams as construction materials is an interesting direction in the development of wooden buildings. An important issue in this case is the strength characteristics of bamboo beams. In this context, the topic of the article should be considered interesting and topical. The introduction and the literature review contained in it are substantively prepared and well introduce the reader to the subject of the article. The cited literature is up-to-date, mostly from the last few years. The work methodology is generally well described. Nevertheless, authors should better characterize the research material. The authors of the study used side-pressure laminated bamboo lumber column. In the literature review, they state that so far research has been conducted rather on the basis of the parallel strand bamboo column. The authors should explain in more detail what is the difference between side-pressure laminated bamboo lumber column and parallel strand bamboo column. Perhaps it would be helpful to take a picture of the cross-section of the tested material.

Response: Thank you very much for your interest and recognition of our work and your precious suggestions. We have revised this paper in accordance with your comments. For the differences between the "parallel strand bamboo" and "side-pressure laminated bamboo" we have given additional explanations in lines 42-48. We also give pictures of both, please refer to Figure 1, Figure 2. Here we provide an item-by-item response to your comments.

Point 1: Why did the authors use only 3 samples for each variant (additionally, a system failure excluded 1 sample from variants B and C). As a standard, in strength tests, approx. 10 samples per variant are used.

Response 1: Thank you very much for your comments, and we also agree with you that this is one of the limitations of this study. Side-pressure laminated bamboo lumber is a modern engineering bamboo material made through specific processes such as gluing and hot pressing, which has less variability than wood and natural bamboo. A study (https://doi.org/10.1016/j.conbuildmat.2017.06.005) shows that a total number of 62 or 30 clear specimens provide similar statistical results. The results of the material property tests (Table 1) also prove a small variability of side-pressure laminated bamboo lumber as its feature. Additionally, there are only three samples per group of specimens in this study, after considering the test cost.

Point 2:  With regard to formula 2, there is no explanation of what the symbol "A" means.

Response 2: Thank you for pointing out the shortcomings of this manuscript. We have added the relevant content as required, please refer to the note of formula 2.

Point 3: The results of experimental research should be confronted by the authors with the data available in the literature. In the entire analysis of the results, there is no reference to the results of studies carried out by other researchers.Taking into account the above remarks, the article requires a minor revision.

Response 3: Thank you for your comments. In this paper, the content of our experimental study is our innovation, which has not been studied by other researchers. Therefore our experimental results have not been analyzed in comparison with the results of other researchers.